# Dynamic Regulation of DNA Methylation and Brain Functions

**DOI:** 10.3390/biology12020152

**Published:** 2023-01-18

**Authors:** Jiaxiang Xie, Leijie Xie, Huixian Wei, Xiao-Jiang Li, Li Lin

**Affiliations:** Guangdong Key Laboratory of Non-Human Primate Research, Guangdong-Hongkong-Macau Institute of CNS Regeneration, Jinan University, Guangzhou 510632, China

**Keywords:** DNA methylation, transcriptional regulation, neurogenesis, development, aging, neurodegenerative diseases

## Abstract

**Simple Summary:**

DNA methylation is involved in biological processes including neurogenesis, aging, and the pathogenesis of brain disorders by the regulation of gene expression. A comprehensive understanding of dynamic DNA methylation changes during development and aging and potential regulatory mechanisms in brain disorders could promote future therapeutic prevention and drug development.

**Abstract:**

DNA cytosine methylation is a principal epigenetic mechanism underlying transcription during development and aging. Growing evidence suggests that DNA methylation plays a critical role in brain function, including neurogenesis, neuronal differentiation, synaptogenesis, learning, and memory. However, the mechanisms underlying aberrant DNA methylation in neurodegenerative diseases remain unclear. In this review, we provide an overview of the contribution of 5-methycytosine (5mC) and 5-hydroxylcytosine (5hmC) to brain development and aging, with a focus on the roles of dynamic 5mC and 5hmC changes in the pathogenesis of neurodegenerative diseases, particularly Alzheimer’s disease (AD), Parkinson’s disease (PD), and Huntington’s disease (HD). Identification of aberrant DNA methylation sites could provide potential candidates for epigenetic-based diagnostic and therapeutic strategies for neurodegenerative diseases.

## 1. Introduction

The brain is the most important organ that serves as the center of the nervous system in mammals. The function of the brain is to control and coordinate a wide variety of actions and reactions, especially thought, memory, and emotion in primates. The development and aging of the brain are complex processes that depend on multiple layers of precise regulation. During aging, the brain shrinks and changes structurally and is predisposed to neurodegenerative diseases. Alzheimer’s disease (AD), Parkinson’s disease (PD), and Huntington’s disease (HD) are common neurodegenerative diseases that cause progressive loss of neuronal function and lead to cognitive impairment. Epigenetic mechanisms play essential roles in brain function by regulating gene expression [1,2,3]. As a critical epigenetic modification, DNA methylation affects transcriptional activity by recruiting or inhibiting the binding of transcription factors to DNA. Many brain disorders display altered gene expression, and accumulating evidence implicates dynamic DNA methylation in altered gene expression and pathological processes in the brain [4,5]. Dysregulation of the epigenome has been reported in neurodevelopmental and neurodegenerative diseases [6].

DNA methylation is a heritable epigenetic mechanism linked to gene expression and the regulation of biological processes [7]. In mammals, 5-methylcytosine (5mC) is formed by DNA methyltransferases (DNMTs) transferring a methyl group from S-adenosylmethionine (SAM) to cytosine at 5-position (Figure 1A). 5mC occurs mainly in the CpG dinucleotides of the mammalian genome. A total of 70–80% of CpGs are modified by 5mC in the human genome [8]. During development and cellular differentiation, the initial establishment of 5mC patterns, termed de novo methylation, relies predominantly on the activity of both DNMT3a and DNMT3b [9], whereas DNMT1 performs a maintenance function over the course of DNA replication to ensure the propagation of 5mC patterns [10].

5mC has been termed the fifth base of the human genome because it plays a key role in inhibiting transcriptional activities (Figure 1B) [11]. A typical situation is the X chromosome inactivation in placental mammals, where 5mC shuts down the transcription of most genes on one of the female’s X chromosomes to compensate for the different dosages of the X chromosome in males and females [12]. However, some studies have reported that several methylation sites are related to gene transcriptional activation [13,14], which may be due to the dynamic binding of transcription factors to certain methylated DNA. 5mC patterns are subject to orderly changes during mammalian development and aging. Neurogenesis proceeds from embryonic stages to the adult brain [15,16]. Dynamic DNA methylation indeed contributes to embryonic and adult neurogenesis, including stem cell maintenance and proliferation, neuronal differentiation and maturation, fate interpretation, and synaptogenesis. Gain or loss of 5mC impacts neuronal development and brain function [17,18]. Growing evidence supports Horvath’s epigenetic clock theory, which is based on 5mC patterns of specific sites across multiple tissues and provides an estimator for predicting chronological age [19]. During aging, an overall genome-wide decreased 5mC (global hypomethylation) has been reported, and regional hypermethylation within the CpG islands of specific gene promoters has been observed [20,21]. Abnormal patterns of 5mC commonly disrupt transcriptional regulation and lead to various diseases, such as cancer and neurodegenerative diseases.

Another family, named Ten-eleven translocation (TET) dioxygenases, is responsible for catalyzing the iterative oxidation of 5mC to 5-hydroxymethylcytosine (5hmC), 5-formylcytosine (5fC), and 5-carboxylcytosine (5caC) on DNA. The TET family in mammals contains Tet1, Tet2, and Tet3, which share a homology catalytic domain at their C terminal. 5fC and 5caC are then specifically recognized by thymine DNA glycosylase to trigger the base excision repair (BER) process, leading to their final conversion to unmethylated cytosine (Figure 1A) [22,23]. Originally, this process was thought to be just an active DNA demethylation with 5hmC, 5fC, and 5caC as transient intermediates. Proteome-wide profiling showed that these modifications may recruit different transcriptional regulators (as readers) to regulate gene expression (Figure 1B) [24,25]. This suggests that novel forms of DNA methylation provide potential regulatory mechanisms for genome functions. Due to the limitations of detection methods for 5fC and 5caC, most studies have focused on the biological functions of 5hmC.

Large differences in 5hmC levels in different tissues have been reported, ranging from 0.03% to 0.69% [26]. 5hmC is typically enriched in active genes, promoters, transcription factor-binding sites, and brain-specific enhancers but is absent in intergenic regions [27,28]. At present, a great number of studies have confirmed that 5hmC generally promotes gene expression. Therefore, 5hmC is considered to be a stable epigenetic marker and is thus called the sixth base of DNA [29,30]. In mammals, 5hmC is the most abundant in the central nervous system (CNS), where mature neurons appear to be the major contributors to 5hmC. 5hmC levels in CNS are approximately 10-fold higher than in embryonic stem cells (ESCs) and 3-fold higher than in peripheral tissues, especially those enriched in Purkinje neurons [26,27,31]. This suggests that high levels of 5hmC are essential for proper neurodevelopment and contribute to the neuropathology of neurodegenerative diseases [5,32].

In the pathogenesis of various neurodegenerative diseases, the silencing or activation of multiple genes has been characterized by hypomethylation or hypermethylation and abnormal regulation of enzymes related to DNA methylation formation or removal. As epigenetic modifications are invertible processes, this provides insights into therapeutic intervention by correcting the aberrant epigenetic status to achieve epigenetic balance. DNA methylation emerges as a promising target for the selection of therapeutic approaches. To date, DNMT inhibitors (DNMTi) are widely used in preclinical and clinical research, among which 5-Azacytidine (5-Aza-CR) and 5-aza-2′-deoxycitidine (5-Aza-CdR) have been approved by the Food and Drug Administration (FDA) in the United States [33,34]. Depending on the interaction of epigenetic modifications, the new therapy has also been tried in clinical applications by combining DNMTi and HDACi (histone deacetylases inhibitors) [35,36,37]. Further understanding of the fundamental mechanisms of epigenetic regulation in neuronal development and diseases could promote therapeutic applications.

In this review, we provide an overview of recent advances in the study of dynamic changes in 5mC and 5hmC during neural and brain development and their abnormalities in neurodegenerative diseases, including AD, PD, and HD (Figure 2). Finally, we discuss potential therapeutic approaches that target DNA methylation mechanisms to alleviate the associated symptoms of neurodegenerative diseases.

## 2. Dynamic Changes in DNA Methylation in Brain Development

5mC plays a prominent role in the normal development of organisms. Abnormal changes in 5mC patterns may lead to developmental disorders, such as intellectual disability, global developmental delays, movement disorders, and growth abnormalities [38]. At early developmental stages, neural stem cells generate all cortical neurons. 5mC alterations at the promoters of genes, such as *Slit1, Bdnf, Wnt3, Esrrb*, and *Tcl1*, that are involved in brain development and neural differentiation may result in transcriptional changes in those genes and, therefore, influence neural differentiation [39]. Cytosine methylation occurs primarily at CpG dinucleotides but is also found at non-CpGs sites. There is a significant association between gene silencing and non-CpG methylation [40]. Notably, non-CpG methylation is a conserved DNA modification in the human neuronal genome that is implicated in the regulation of genes related to neuronal differentiation, synaptogenesis, and function [38]. Unlike CpG-methylation, non-CpG methylation is established after the maturation of neurons in human and mouse brains. In humans, 5mC levels at certain loci that are responsible for the growth and development of CNS increase significantly after birth and participate in neuronal differentiation—a dramatic change that can also persist throughout the whole life [41,42]. Interestingly, mature neurons never undergo mitosis, which means that although 5mC cannot be passively eliminated during cell division, it can still be actively abolished by TETs [43]. Therefore, 5mC is dynamic and plays a role in neurogenesis. Similarly, DNA methylation is involved in the regulation of astrocyte differentiation and maturation. A potential impact on sexual differentiation in the developing mammalian brain has been implicated, in which 5mC is required for the masculinization or feminization of the brain [44,45].

The dynamics of 5mC depend on a series of enzymes, including DNMT1 for methylation maintenance and DNMT3a/b for de novo methylation. DNMT1 is a highly conserved DNA methyltransferase in mice and humans that maintains the DNA methylation pattern faithfully during cell division [46]. In mice, DNMT3a is highly expressed in neural stem and progenitor cells in the developing cerebral cortex and neurons and oligodendrocytes (but with very low expression in astrocytes) [47,48]. DNMT3b is expressed only in the ventricular zone during the early stages of embryonic development [17]. Mice deleted of *Dnmt3a* (*Dnmt3a-/-*) can survive for four weeks, but *Dnmt1* or *Dnmt3b* homozygous deletion (*Dnmt1-/-*, *Dnmt3b-/-*) resulted in embryonic or postnatal lethality [9,49]. Conditional mutant mice that lack *Dnmt1* or *Dnmt3a* in forebrain excitatory neurons exhibited learning and memory impairment and deregulated gene expression in neurons. These findings revealed the roles of *Dnmt1* and *Dnmt3a* in maintaining DNA methylation and modulating neuronal gene expression in adult neurons [50]. Furthermore, *Dnmt3b* is essential for de novo methylation during development, and a lack of *Dnmt3b* led to abnormal development of the rostral neural tube [9].

5hmC mappings of mice and humans showed that genome-wide 5hmC levels in brain tissues are significantly higher than in other tissues, reaching more than 10-fold [51,52,53]. 5hmC levels are increased markedly from the early postnatal period until adulthood, suggesting a strong correlation between elevated 5hmC and expression levels of genes important for normal neural development and activity [28,53]. *Tet1* and *Tet2* are highly expressed in embryonic stem cells ESCs [51]. However, Tet3 expression levels were barely detectable in ESCs but rapidly increased during neuronal differentiation [54]. Several studies have shown that the regulation of the expression of 5hmC and TET family enzymes is important for neurogenesis and neurodevelopment. *Tet1/2/3* triple knockout in ESCs compromised neural differentiation and manifested as developmental defects during gastrulation in early mouse embryos [55,56]. The loss of *Tet2* contributed to enhancer hypomethylation by oxidizing 5mC and delayed the timing of transcriptome reprogramming, which influenced the differentiation of ESCs to neural progenitor cells (NPCs) [57]. In *Tet3*-knockdown of NPCs, a dramatic genome-wide loss of 5hmC has been reported, which led to a de-repressing of pluripotency-associated genes such as *Oct4* and *Nanog*. These findings indicated that *Tet3* plays a critical role in NPC specification, maintenance, and terminal differentiation [39,54,58].

During neurodevelopmental stages, *Tet1* deficiency resulted in stage-dependent defects during the transition from NPCs to oligodendrocyte progenitor cells (OPCs), which, in turn, affected the formation of mature myelinating oligodendrocytes (OLs) and remyelination in the mouse brain [52,59]. The inhibition of *Tet1/Tet3* impaired the branching of GCs dendrites and gene expression related to axon guidance, dendrite outgrowth, and ion channel functions [26,60]. *Tet3* was required for regulating excitatory and inhibitory synaptic transmission [61]. In clinical observations, human TET3 deficiency led to the common phenotypic features of intellectual disability and/or overall developmental delays [62]. 5hmC was preferentially enriched in genes related to synapse-associated functions in human and mouse brains [63]. Genome-wide 5hmC profiling data revealed a unique pattern in the cerebellum during brain development in rhesus monkeys [64]. It has been demonstrated that 5hmC levels increase sharply during cerebellar granule cell (GC) development, which is consistent with the accumulation of 5hmC in the cerebellum [60]. Taken together, all the above-mentioned studies suggest that 5hmC is involved in the epigenetic regulation of normal physiological function and neuronal circuits in the cerebellum.

In conclusion, it is clear that the establishment of and dynamic changes in 5mC/5hmC are involved in neuronal development and differentiation in a cell- and region-specific manner, evidenced by the inactivation of methylation-associated enzymes that can result in developmental anomalies and lethality.

## 3. DNA Methylation Is Involved in Brain Aging

Aging is an inevitable process in organisms, beginning with embryonic stem cells that gradually lose molecular fidelity during maturation, leading to structural degradation, functional decline, and, ultimately, disease and death. A growing body of evidence links aging to changes in epigenetic factors, including DNA methylation, histone modifications, and telomere shortening. They collaborate to play an important role in regulating the aging process (Figure 3). Various animal models, such as fly, killifish, and mouse models, are essential to uncovering the mechanisms associated with aging [65]. Biomarkers to predict aging have been developed, e.g., epigenetic clocks, telomere length, transcriptomic predictors, proteomic predictors, metabolomics-based predictors, and composite biomarker predictors [66]. In recent years, 5mC has gradually gained attention as a reliable biomarker to evaluate aging. Numerous studies have found that age-associated DNA methylome changes in certain genomic regions occur across species. 5mC at certain CpG sites is highly correlated with age; thus, using a relatively small number of CpG sites can accurately predict biological age. Based on this, such modifications can be used as an estimator of aging in organisms, termed the epigenetic clock [19,67,68,69]. The epigenetic clock allows us to accurately predict the aging process across a lifespan based on the 5mC status of specific sites. A representative example is that CpG sites adjacent to ELOVL Fatty Acid Elongase 2 (*ELVOL2*) exhibited significant hypermethylation with age; thus, *ELOVL2* has been used as a typical epigenetic clock in humans and mice [70]. Notably, the epigenetic clock describes biological aging, not chronological age.

An unresolved question is whether DNA methylation functions as a footprint mark or driver of aging. A number of studies have explored the relationship between DNA methylation and aging. Currently, DNA methylation is regarded as a “ruler” measuring the aging of organisms. Mammalian cells undergo global DNA hypomethylation (enriched in CpG islands) and local DNA hypermethylation (at gene-poor, late-replicating, lamina-associated domains) with aging [21,71]. A recent finding revealed that local high/low CpG density loci displayed distinct trajectories in age-related methylation dynamics. CpG density in the genome has a major influence on both the local tick rate of age-associated DNA methylation changes and inter-individual variability [72]. Epigenetic aging rates based on DNA methylation were significantly correlated with race/ethnicity. Analyses of multiple tissues of different racial/ethnic groups suggested that habitat may be a noticeable factor affecting the acceleration of aging [73]. Interestingly, the epigenetic clock of DNA methylation is also correlated with gender. Young males exhibited higher epigenetic aging rates than young women, which likely continue into old age, especially in brain tissue [74,75,76]. However, the cerebellum ages more slowly, and shows no significant sex-specific effects [73]. Accumulation of DNA damage and DNA repair mechanisms are associated with aging and age-related genomic instability in males; otherwise stated, the efficacy of DNA repair mechanisms may differ between males and females [77,78]. Another explanation is that removing androgen could decelerate the rate of aging [79]. However, this difference between males and females has been difficult to verify in animal models. Therefore, it is necessary to analyze the differences in the epigenetic aging rates of males and females throughout their lifespans.

Szulwach et al. showed that 5hmC levels were significantly increased in the cerebellum and hippocampus of 6-week-old and 1-year-old adult mice compared with 7-day-old mice, suggesting that 5hmC may be directly involved in the maturation and aging of mouse brains [27]. Using freshly isolated brain tissues from rhesus monkeys, they found that 5hmC accumulated in four brain regions (cortex, cerebellum, hippocampus, and striatum) as age increased [64]. This result suggested that dynamic 5hmC is involved in the aging process of primate brains and may play a role in the pathogenesis of aging-related brain diseases.

In summary, differential DNA methylation, including 5mC and 5hmC, has been implicated in the complex biological process of aging. Along with 5mC, which was confirmed as one of the most promising predictive biomarkers of aging, we speculate that 5hmC may be another potential candidate for the epigenetic clock. The accumulation of data regarding 5hmC profiling in multiple tissues from different ages and across species will help us gain insights into aging mechanisms and design treatments aimed at slowing the process of aging.

## 4. Alterations in 5mC/5hmC and Methylases in AD

AD is a clinically neurodegenerative disease characterized by the neuritic plaques resulting from the accumulation of amyloid-beta peptide (Aβ) and neurofibrillary tangles formed by hyperphosphorylated tau in the brain—the histopathological hallmarks that occur early in the hippocampus and entorhinal cortex and spread to other brain regions as the disease progresses [80]. AD usually manifests as declines in memory, cognitive function, abstract thinking, and judgment, in addition to changes in behavior, mood, and emotions [81,82]. Aging and sex are the two major risk factors that influence epigenetic patterns in mammalian tissues of AD [83,84]

Alterations in 5mC and 5hmC were found in AD patients and mouse models, and DNA methylation was reportedly positively correlated with AD markers including amyloid beta, tau, and ubiquitin loads [85]. Common hypomethylation was found at the gene enhancers in AD neurons in the entorhinal cortex, hippocampus, dorsolateral prefrontal cortex, and cerebellum [86,87,88,89]. A genome-wide reduction in 5hmC was found due to decreased TET enzymatic activities in neurons but not in astrocytes in AD mouse models and AD patients’ brains [90]. However, several studies have shown that 5mC and 5hmC are significantly increased in the middle frontal gyrus (MFG), middle temporal gyrus (MTG), frontal cortex, and hippocampus of AD patients. These discrepancies may be caused by differences in sampling and analytical methods. However, what is certain is that alterations in genomic DNA methylation are involved in the pathological process of AD [85,87].

Presenilin1 (PSEN1) is a key enzyme that cleaves the amyloid-β protein precursor (AβPP) to generate the amyloid-β (Aβ) peptides associated with AD. Hypomethylation of 5mC at *PSEN1* has been linked to its elevated expression in AD brain cells [91,92]. AD is also affected by 5mC variants in multiple loci, including *SORL1, ABCA7, HLA-DRB5, SLC24A4,* and *BIN1*, whose protein products have been reported to be present in both Aβ and tau tangle aggregates [93].

The aberrant expression of TET enzymes has been related to the pathogenesis of AD. In the hippocampus of aged transgenic AD mice, a considerable decline in 5hmC and Tet2 levels was observed, which correlated with marked Aβ plaque accumulation, GFAP-positive astrogliosis, Iba1-positive microglia overgrowth, and the overproduction of pro-inflammatory factors. After restoring *Tet2* expression in adult neural stem cells isolated from the hippocampus of aged transgenic mice, 5hmC levels could be rescued and mice demonstrated enhanced regenerative capacity, suggesting that *Tet2* might be an intriguing target for brain regeneration during aging and in AD [94]. In the hippocampus of *APP/PSEN1* double-transgenic mice, knockdown of *Tet3* blocked neuronal differentiation and increased the astrocytic differentiation of NSCs, indicating that TET3 proteins may positively regulate the neuronal differentiation of NSCs and cognitive function in AD mouse models [95,96].

Formaldehyde is the main precursor of most complex organic molecules [97]. Increased concentrations of formaldehyde in the brain can lead to neurological dysfunction and degeneration [98,99]. The process of DNA demethylation is accompanied by the generation of formaldehyde [100]. Early AD-like changes occurred in mice exposed to excessive formaldehyde daily for one week [101]. Following chronic intracerebroventricular injection of formaldehyde in rhesus monkeys, the animals exhibited Aβ neuritic-like plaques, neurofibrillary tangle-like structures, increases in phosphorylation tau proteins, and neuronal loss in the hippocampus, entorhinal cortex, and prefrontal cortex [102]. Interestingly, an active DNA demethylation process was also induced upon formaldehyde administration [103]. The injection of excessive formaldehyde into the hippocampus of healthy adult rats mimics abnormalities in learning and cognition. This may be due to the reduction in global DNA methylation levels caused by the inhibition of DNMT1 and DNMT3a activity through the modification of their cystine residues by excess formaldehyde [104]. Excessive physiological levels of formaldehyde interfered with overall DNA methylation reduction in AD patients [104]. Therefore, excessive formaldehyde exposure from the environment led to abnormal DNA methylation and may be a risk factor for AD. Formaldehyde scavengers may be an effective method for the treatment of AD [105] (Figure 4).

DNA methylation editing has been used as a novel treatment for a variety of diseases. Abnormal CpG methylation of the *APP* led to abnormal gene expression of the *APP* in the brains of AD patients. *APP* hypermethylation can be achieved by the editing system to fuse Tet1 or Dnmt3a with dCas9 to effectively inhibit *APP* in primary neurons and in vivo mouse brains [106,107,108].

Taken together, in AD animal models or patients, altered DNA methylation and associated risk factors have been identified, which might influence the onset and progression of AD. Since epigenetic modification is a dynamic and reversible process, it is worthwhile to explore novel AD treatments by manipulating DNA methylation.

## 5. Regulation of DNA Methylation in PD

PD is characterized by neuronal death in particular regions of the substantia nigra and widespread accumulation of an intracellular protein, alpha-synuclein (SNCA) [109]. The disease causes the excessive loss of dopaminergic neurons before the appearance of motor symptoms [110]. Aberrant accumulation of a-synuclein in the cytoplasm of certain types of neurons forms aggregates, which comprise the majority of Lewy bodies and are observed in neocortical brain regions, cholinergic and monoaminergic brainstem neurons, and neurons in the olfactory system [111].

Epigenetic mechanisms, particularly dysregulation of DNA methylation, have been linked to PD. Genome-wide profiling of 5mC in brain and blood samples from PD patients revealed coordinated variations in 5mC associated with PD pathology [112]. Since SNCA is the most critical factor in the pathogenesis of PD, a growing body of studies has focused on the regulation of DNA methylation on SNCA. Mastumoto et al. reported a regional non-specific 5mC reduction in *SNCA* intron 1 in the substantia nigra of PD patients. Consistently, *SNCA* expression is increased along with altered 5mC status [113]. Another study also demonstrated that 5mC changed in SNCA intron 1 in the brains of PD and early-onset Parkinson’s disease (EOPD) patients [114,115,116]. Another family member, β-synuclein (*SNCB*), has been found to directly interact with *SNCA*, modulating its activity and limiting its oligomerization in vitro and in vivo [117]. In the frontal and temporal cortices of PD patients, hypomethylation of 5mC was found at the *SNCB* promoter and associated with a significant decrease in *SNCB* expression [118]. All the above studies suggested that 5mC dysregulation at the synuclein loci contributed to PD pathogenesis.

Besides synuclein, the methylation levels of other genes have also been reported to show differences in PD patients. Kwok et al. investigated the transcriptional activity of the microtubule-associated protein Tau (*MAPT*) and proved that increased MAPT is a vulnerable feature in idiopathic PD [119]. The subsequent study revealed that age at onset was positively correlated with 5mC and *MAPT* methylation in leukocytes from PD samples. However, hypermethylation of 5mC at the *MAPT* gene is neuroprotective by reducing MAPT expression [120]. Increased expression of inflammatory cytokine tumor necrosis factor α (*TNF-α*) has been associated with the death of dopaminergic cells in PD [121]. 5mC status at the CpG site of the *TNF-*α promoter near or within transcription factor-binding sites regulated the binding of the transcription factors *AP-2* and *Sp1*, further impacting TNF-a transcriptional activation [122]. In addition, many other genes showed dysregulation of 5mC, including *FANCC* cg14115740, *TNKS2* cg11963436, *PGC-1α*, and *NOS2* by different screenings in PD cases [123,124,125,126].

The effects of DNMTs on the pathogenesis of PD were revealed through the study of DNMT inhibitors. In dopaminergic neuronal cells, pretreatment with the DNMT inhibitor 5-aza-CdR increased the neurotoxic damage to dopaminergic neurons. DNMT inhibitor-induced demethylation may accelerate dopaminergic neuron death by rendering them more susceptible to neurotoxins and erroneously affecting the transcription of key PD-related genes [127].

In an induced cell model of PD, genome-wide mapping showed an aberrant 5hmC landscape that is correlated with cell cycle related-genes. In the substantia nigra pars compacta of MPTP-induced PD mice, downregulation of TET2 attenuated dopaminergic neuronal injury, whereas MPTP-induced motor deficits were found [128]. Differentially hydroxymethylated regions (DhMRs) of 5hmC were enriched in the genes associated with neurogenesis and neuronal development in the substantia nigra of PD patients [129]. However, there are also different views. Kaut et al. found changes in 5hmC in different brain regions of PD patients [130]. In the cerebellum, 5hmC was significantly upregulated, but the levels of both 5mC and 5hmC in the brain stem and substantia nigra and the dopaminergic neurons were unaltered. Stöger et al. also revealed that PD patients’ cerebellums have higher 5hmC levels and that alterations are independent of gender [131]. This controversy suggests that there is indeed a dynamic 5hmC pattern in the brain of PD, and further research is needed to explore the underlying mechanisms.

Based on the regulatory role of DNA methylation concerning genes related to PD pathogenesis, identifying epigenetic targets as a therapeutic strategy for the treatment of PD has become a research hotspot [132,133]. It has been reported that dopamine replacement therapy with 3, 4-dihydroxy-l-phenylalanine (L-DOPA) is effective for motor symptoms of PD [134]. In PD patients, L-DOPA treatment enhanced both in vivo and in vitro a-synuclein DNA methylation [135] (Figure 5).

A novel plant-derived biopharmaceutical compound, atremorine, reduced the production of tyrosine hydroxylase in the substantia nigra, blocked microglia activation, and prevented neuronal death. It has a neuroprotective effect on the dopaminergic neurons in the substantia nigra, possibly by improving DNA methylation and exerting epigenetic regulation [136]. For example, atremorine prevented 1-methyl-4-phenyl-1,2,3,6-tetrahydropyridine (MPTP) from inducing dopaminergic neurodegeneration in the substantia nigra and microglia activation and neurotoxicity in the nigra-striatal region. Mice with MPTP-induced neurodegeneration also had improved motor and cognition function [136] (Figure 5). Therefore, this drug has been considered a potential dopamine enhancer, acting as a neuroprotective agent with possible prophylactic and therapeutic activity in PD [137].

Here, we summarized the current understanding of the regulation of DNA methylation in PD-associated genes, which is only a preliminary insight into the epigenetic complexity of PD pathology. With better-characterized animal models, sufficiently large patient cohorts, and adequate bioinformatic analysis methods, we will gain more insights into the underlying mechanisms of neurological disorders in PD in the future.

## 6. Aberrant 5mC and 5hmC in HD

HD is an autosomal dominant neurodegenerative disease caused by the progressive loss of neurons in the human brain due to an abnormal expansion of CAG trinucleotide repeats within the coding exon 1 of the huntingtin (*HTT*) gene [138]. It becomes pathogenic when CAG repeats expansion exceeds more than 35. The length of the expanded repeats is associated with the age of onset and severity of the disease [139,140]. Mutant *HTT* produces a protein with an abnormally elongated polyglutamine (polyQ) tract at its N-terminus that exhibits misfolding and is prone to oligomerization and aggregation. Mutant *HTT* affects multiple aspects of neuronal function, including mitochondrial dysfunction, synaptic dysfunction, impaired protein homeostasis, and defective vesicular trafficking [141]. Expression of the mutant proteins triggers pathological changes in the brain, with progressive loss of selective populations of neurons in the striatum and cortex, leading to abnormal involuntary movements, cognitive decline, and a range of psychiatric disorders [142].

There is growing evidence that genome-wide alterations in DNA methylation are strongly associated with HD, suggesting that epigenetic abnormalities and changes in the genomic levels in patients may be possible mechanisms underlying the pathogenesis. Linking the pathogenesis of HD to altered epigenetic modifications may lead to the potential development of epigenetic-based HD treatment or intervention [143,144].

Based on the epigenetic clock from different brain regions of 26 HD cases and 39 controls, a significant epigenetic effect of aging has been observed in specific brain regions. HD patients increased biological age for 3.2 years and displayed disrupted 5mC levels in the brain [145]. Zadel et al. explored the presence of specific 5mC patterns in the blood samples of HD patients and concluded that 5mC in the blood was insufficient to be a viable biomarker for predicting HD as the change was not statistically significant [146]. However, in 2020, another study came to a completely different conclusion [147]. An epigenome-wide association study (EWAS) of human blood samples revealed that HTT mutation status was significantly associated with 33 CpG loci, including the *HTT* gene. The latter conclusion was supported by other studies of Q175 HTT knock-in mouse and transgenic sheep models [147].

By mapping 5mC loci in striatal cells carrying polyQ-expanded HTT, many genes were found to be significantly altered in 5mC with mutant *HTT* [143]. By analyzing 5mC levels in cortical neurons and the livers of HD patients, 38 differentially methylated binding sites at the *HTT* gene were identified, with higher *HTT* expression observed in the cerebral cortex [148]. The authors further revealed that 5mC levels may be associated with the age of disease onset in the cortex. From 136 differentially enriched loci associated with neuronal development and neurodegeneration in HD cases, hypermethylation of 5mC at the *HES4* promoter was found to correlate with loss of histone methylation (H3K4me3), striatal degeneration, and age at onset [149]. Jia et al. also observed increased 5mC in Lys (K)-specific demethylase 5D (KDM5D) in normal and HD cells treated with histone deacetylase (HDAC) inhibitors. Further, they demonstrated that the HDAC inhibition-treated male offspring of HD transgenic mice exhibited a significantly improved HD phenotype, which was associated with increased expression of Kdm5d [150]. These findings revealed that 5mC was broadly changed in HD and suggested that interactions of different epigenetic mechanisms may contribute to HD phenotypes.

In the putamen of HD patients, increased levels of 5mC and decreased levels of 5hmC were found at the 5′UTR region of the ADORA2A gene, which encodes a G-protein-coupled receptor, Adenosine A2A receptor (A2AR). The A2AR protein was severely downregulated in HD patients and mouse models [151]. This suggested that 5mC and 5hmC may coordinate gene expression in disease pathology. Interestingly, a genome-wide loss of 5hmC has been reported in YAC128 (128 CAG repeats) HD mouse striatum and cortex. Among 747 differentially hydroxymethylated regions (DhMRs) in the striatum, the gain of DhMRs in the genome was positively correlated with gene transcription. The DhMRs-annotated genes were associated with neuronal development, differentiation, function, and survival [152].

DNA methylation and related methylases have been implicated in the diagnosis and treatment of HD. The latest study in 2022 found that 237 selected CpGs could be used for accurately distinguishing HD patients from healthy controls by artificial neural network techniques [153]. It has been reported that knockdown of DNMT3A or DNMT1 protected neurons from mutant HTT-induced toxicity, and DNMT was required to mediate neuronal death triggered by mutant HTT. Inhibitors of DNMT, decitabine, and FdCyd blocked mutant HTT-induced toxicity in primary cortical and striatal neurons [144,154] (Figure 6).

In conclusion, recent reports regarding aberrant 5mC and 5hmC in HD suggested a potential impact of DNA methylation on neurogenesis and cognitive decline in HD pathology in a complex manner. Specific CpG sites and DNA transferases may be potential therapeutic targets for HD therapy.

## 7. Crosstalk of DNA Methylation and Histone Modifications

Besides DNA methylation, histone modification is also an important epigenetic mechanism, including histone acetylation, methylation, and ubiquitination. Different epigenetic modifications are interrelated and often interact with each other [155,156,157]. Although the accurate molecular mechanism of the interplay among epigenetic modifications is not well understood, accumulating evidence suggests that epigenetic crosstalk among different epigenetic modifications is involved in aberrant processes of gene transcription and disease development. In 2008, whole-genome-wide CpG methylation profiling indicated a correlation between DNA methylation and histone methylation, including both positive (H3K9me) and negative (H3K4me) correlations [158]. In subsequent studies, certain regulatory domains, including DNMT1-UHRF1 (ubiquitin-like, containing PHD and RING finger domains) [159,160,161] and ADD (ATRX-DNMT3-DNMT3L) domains, which have specific DNA methylation patterns, were found to guide histone modification during gene silencing [162,163]. Conversely, chromatin modifications specify DNA methylation patterns [164]. For instance, H3K9 methylation recruits DNMT3B to its readers, HP1α and HP1β [165]. The knockout of methyltransferases of H3K9me abolishes the localization of DNMT3B and DNA methylation. The interplay of DNA methylation and histone modifications has been revealed in disease development. Tatton-Brown-Rahman syndrome (TBRS) is a developmental disorder with macrocephaly and intellectual disability. In TBRS, mutant DNMT3A impaired the DNMT3A-H3K36me2/H3K36me3 interaction, which led to 5mC hypermethylation in crucial genes enriched with H3K27me3 [166,167]. During tumorigenesis, both aberrant DNA methylation and histone acetylation may cause gene mutations in patients, which eventually lead to gene silencing. Therefore, it is necessary to uncover the precise mechanisms of gene expression and silencing during normal development and disease progression by integrating different epigenetic modifications in future studies.

## 8. Present Challenges and Perspectives

Over the past few decades, remarkable progress has demonstrated that the epigenetic regulation of genome functions has critical roles in brain development and aging. Dynamic and ordered 5mC and 5hmC changes are required to maintain normal biological processes. The disruption of DNA methylation in the brain contributes to neurodegenerative diseases. However, precise mechanisms by which DNA methylation regulates gene expression to contribute to brain development and aging and disease pathology are not fully understood. Additionally, there are some challenges to further exploration. First, the brain is the most complex organ and is divided into multiple regions, each composed of different types of cells. Moreover, different regions of the brain are affected to different extents in neurodegenerative diseases. It is widely accepted that DNA methylation patterns are specific to different brain regions and cell types [64,168]. Therefore, it is difficult for most studies to draw a unified conclusion on the outcome of the regulation of DNA methylation. Second, the relationship between DNA methylation and gene expression is not a simple linear correlation. It has been reported that epigenetic factors could interact with each other or cooperate in a complex manner in the regulation of gene expression [169,170]. A better delineation of the crosstalk between different modifications is necessary to fully understand the epigenetic mechanisms underlying brain physiology and pathology. Third, many epigenetic analyses in humans are based on whole blood and postmortem tissues. It is difficult to obtain fresh tissues from the human brain in a timely manner to examine DNA methylation at different stages of development and aging, especially at the early stages of brain diseases. The DNA methylation of blood does not necessarily represent the bona fide state of the brain. As we know, the epigenetic state is sensitive to environmental conditions. Uncontrolled premortem and postmortem conditions affect the quality of collected samples. From this point of view, non-human primates become an ideal animal model for investigating epigenetics based on their close similarity to humans.

In the future, the coordinated investigation of DNA methylation using appropriate animal models, simultaneously integrating other epigenetic modifications, and more careful consideration of the differences among cell types and brain regions will provide novel perspectives and insights for understanding the dynamics of DNA methylation in brain development, aging, and diseases. We hope such insights will eventually help identify potential targets for diagnostic and therapeutic strategies for human brain diseases and develop new epigenetic therapeutic drugs.

## 9. Conclusions

DNA methylation is dynamically regulated by different methyltransferases and demethylases. This dynamic applies to a variety of biological processes, ranging from cell differentiation, organ development, and aging responses. Aberrant DNA methylation affects genome instability and altered gene expression, resulting in brain disorders. This review provides a comprehensive overview regarding the contribution of 5mC and 5hmC to normal development, aging, and the pathogenesis of neurodegenerative diseases. Furthermore, we summarize potential biomarker candidates for neurodegenerative diseases, focusing on the genes with altered DNA methylation. Research on the molecular mechanisms involved in the relationship between DNA methylation and the alteration of brain function is still limited. Recent technological advances provide new promising opportunities to explore the precise mechanisms of how DNA methylation regulates aging and diseases, including profiling pervasive and stable changes in DNA methylation, discovering specific enzymes for candidate therapeutic strategies, and linking specific histone modifications to specific signaling pathways. A better understanding of the critical roles of epigenetics during aging and disease processes will lead to improved health and lifespans.

## Figures and Tables

**Figure 1 biology-12-00152-f001:**
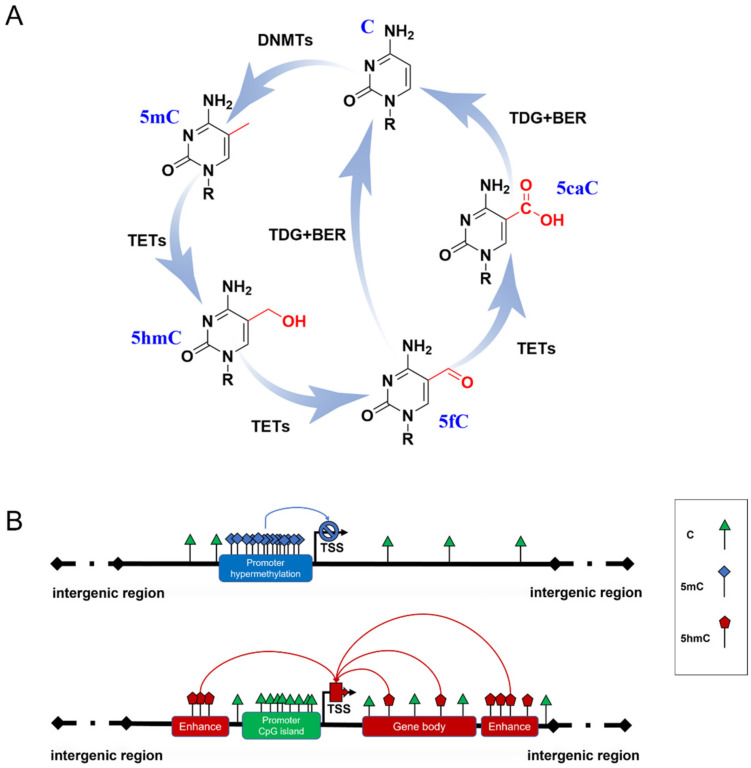
DNA methylation formation and regulation in gene expression. (**A**) The cycle of DNA methylation. 5mC is formed from C by the catalyzation of DNMTs; then, it can be hydroxylated to 5hmC and further oxidized to 5fC and 5CaC by TET enzymes. 5fC and 5caC are recognized by thymine DNA glycosylase (TDG), which triggers the base excision repair (BER) process, leading to the conversion to unmethylated cytosine. (**B**) Possible role of 5mC and 5hmC in gene expression. Hypermethylated 5mC at CpG islands of gene promoters generally inhibits gene transcription (blue line). 5hmC of gene enhancers and gene bodies up-regulates gene expression (red line).

**Figure 2 biology-12-00152-f002:**
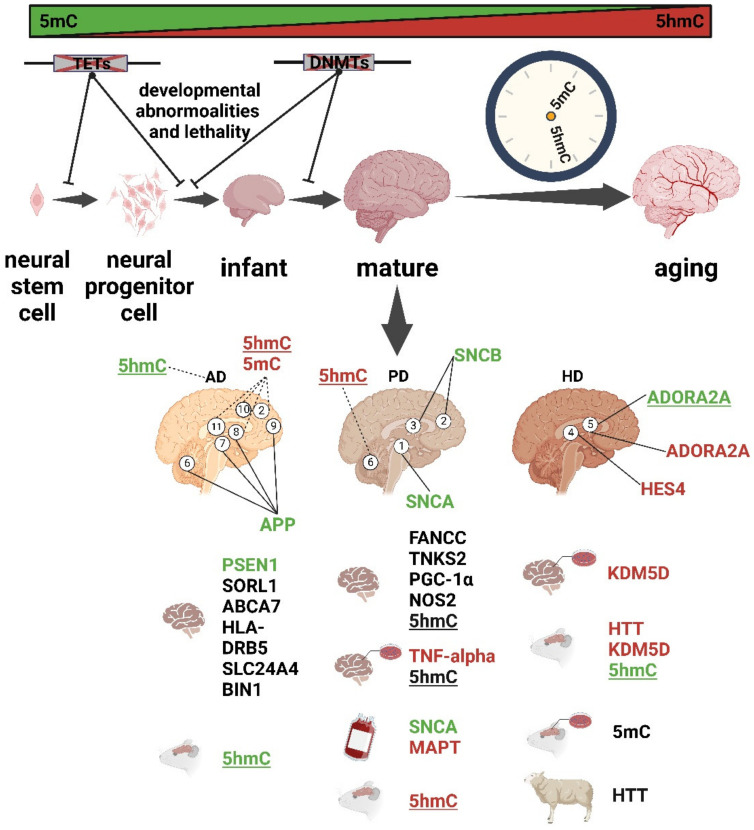
Summary of dynamic 5mC and 5hmC in development and aging and their abnormalities in neurodegenerative diseases (AD, PD, and HD). Green, hypomethylation; red, hypermethylation; dashed line, the conclusion is controversial. Different brain regions are numbered as follows: 1. substantia nigra, 2. frontal cortices, 3. temporal cortices, 4. striatum, 5. putamen, 6. cerebellum, 7. entorhinal cortex, 8. hippocampus, 9. dorsolateral prefrontal cortex, 10. middle frontal gyrus (MFG), and 11. middle temporal gyrus (MTG). Not underlined, global 5mC change; underlined, global 5hmC change.

**Figure 3 biology-12-00152-f003:**
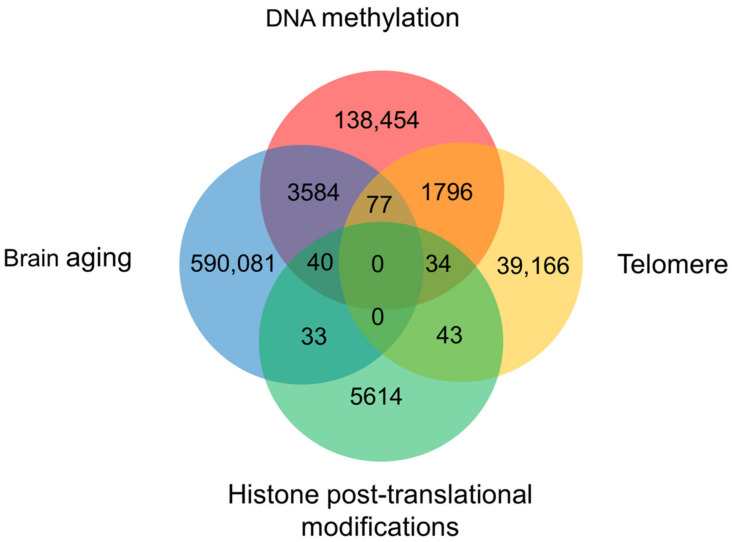
The weight of DNA methylation, histone modifications, and telomere in the process of brain aging. A Venn diagram created by summarizing publications in the Web of Science Core Collection, Chinese Science Citation Database, KCI-Korean Journal Database, MEDLINE, and SciELO Citation Index since 1950.

**Figure 4 biology-12-00152-f004:**
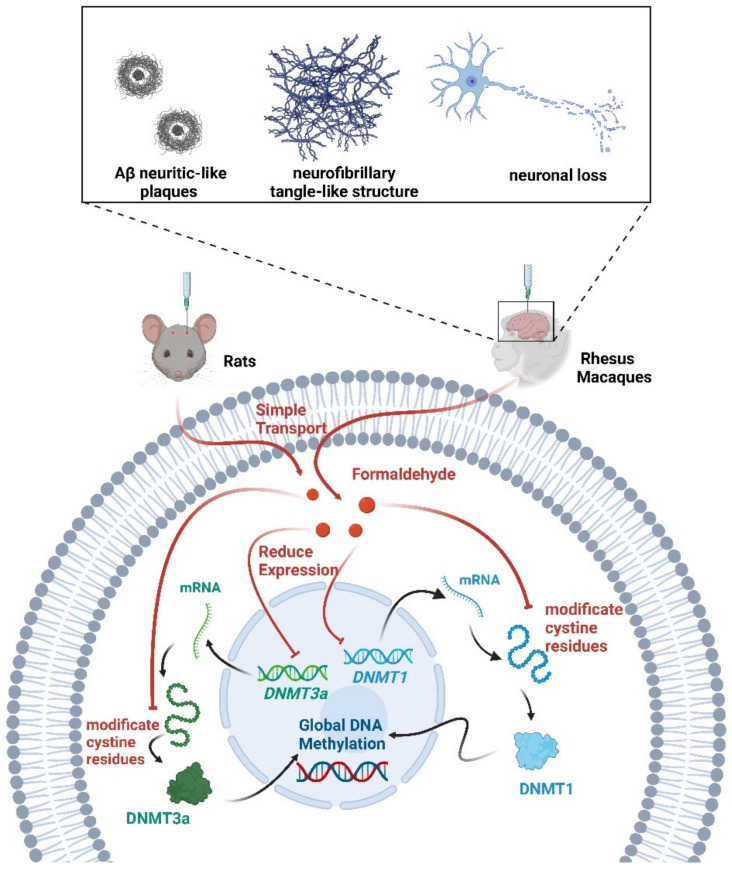
Multiple roles of formaldehyde in methylation. Formaldehyde decreased *DNMT1/3a* expression and global DNA methylation levels. Formaldehyde also inhibited DNMT1/3a activity by modifying cystine residues. Following the injection of formaldehyde into the brains of rats and rhesus monkeys, the animals exhibited symptoms associated with AD.

**Figure 5 biology-12-00152-f005:**
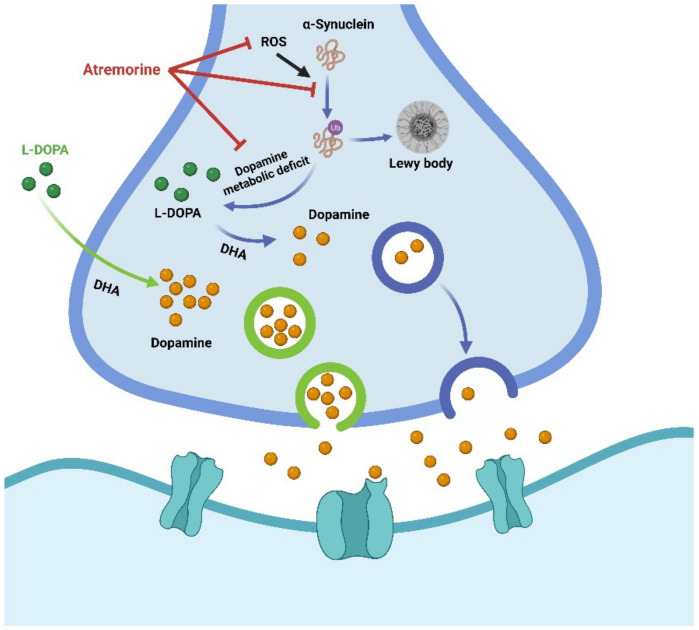
Schematic diagram of L-DOPA and atremorine against MPTP-induced dopaminergic neurodegeneration. Atremorine has an antioxidant impact, including inhibition of reactive oxygen species (ROS), restoration of total ROS levels, and a significant improvement in total antioxidant capacity. Atremorine inhibits the ubiquitin-proteasome system and accumulation of α-synuclein to regulate dopamine/L-Dopa levels in the brain and exhibits neuroprotective effects. Dopamine levels can also be regulated by direct supplementation with L-DOPA.

**Figure 6 biology-12-00152-f006:**
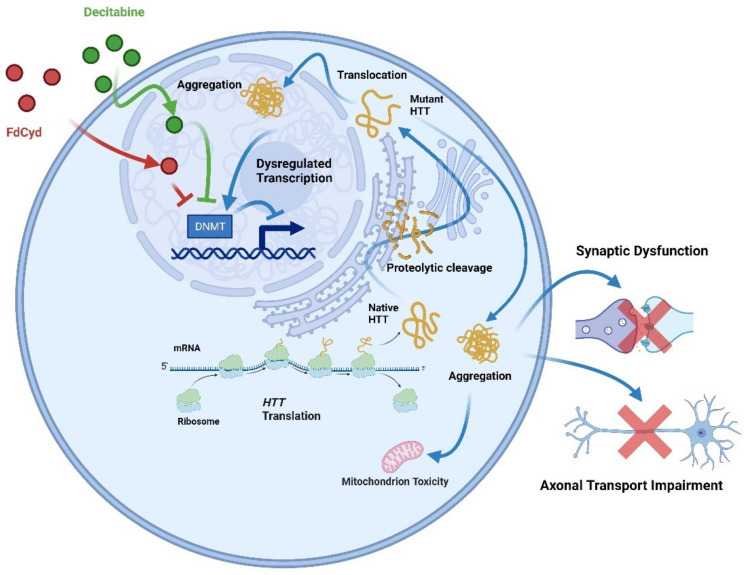
Inhibition of DNMTs (decitabine or FdCyd) in HD neurons blocks mutant Htt-induced transcriptional repression. The full-length huntingtin is produced by translating *HTT*. Proteolysis cleaves native full-length huntingtin to produce extra protein fragments, which form aggregates in the nucleus and cytoplasm. Huntingtin aggregation causes global cellular impairments, such as synaptic dysfunction, mitochondrial toxicity, and decreased axonal transport. Decitabine or FdCyd, as an inhibitor of DNMTs when administered to HD neurons, prevents the transcriptional repression induced by mutant HTT.

## Data Availability

Not applicable.

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
