# Peer review of "Dynamic Regulation of DNA Methylation and Brain Functions"

_biology, 2023, doi:10.3390/biology12020152_

Round 1
Reviewer 1 Report
This review by Xie et al. provides a summary review of DNA methylation in brain disorders and ageing. The manuscript is a focused review of published work addressing methylation changes in specific genes. I am not sure if data is available on global methylation changes in specific regions of the brain in the specific disorders described. This would be an important addition to the review. It may also explain specific regions of the brain involved in the pathology. For example, in PD, what are the regional changes in methylation? Are there changes between PD with Lewy body dementia and without, to explain the differences in the pathology and severity of the disease.
Author Response
Reviewer 1
Comments and Suggestions for Authors:
This review by Xie et al. provides a summary review of DNA methylation in brain disorders and ageing. The manuscript is a focused review of published work addressing methylation changes in specific genes. I am not sure if data is available on global methylation changes in specific regions of the brain in the specific disorders described. This would be an important addition to the review. It may also explain specific regions of the brain involved in the pathology. For example, in PD, what are the regional changes in methylation? Are there changes between PD with Lewy body dementia and without, to explain the differences in the pathology and severity of the disease.
Author Response: We thank the reviewer’s suggestion. There are limited studies regarding global methylation changes in AD and PD brain, which were cited in Page 8, lines 284-287, reference 85 for AD; and Page 10, line 353-355, reference 112 for PD. We also summarized the specific regions of global methylation changes in Figure 2. We were unable to find out studies regarding the correlation between DNA methylation and Lewy body dementia, which is an interesting project in the future studies. We hope that our review will draw more attention to the relationship between regional DNA methylation and selective neurodegeneration in different neurodegenerative diseases.

Reviewer 2 Report
Comments on biology-2119826
In this study, the author has studied “Dynamic regulation of DNA methylation and brain functions.” A lot of studies have already been carried out on a similar topic, and comprehensive data is available in the literature. Sentence-making is weak in this manuscript. The English language used in the manuscript needs major improvements as there are some punctuation and grammatical mistakes present throughout the manuscript. Experimental designs required more clarity. Moreover, research models are not discussed in an understandable manner. Repetition of lines is common, which reflects that the author needs a more comprehensive way of thinking.
Specific comments:
1. Page 1, line 14-15: “Alzheimer’s disease (AD), Parkinson’s disease (AD) and Huntington’s Disease (HD).” Please remain consistent; either capitalize the word ‘D’ in all diseases or small the word ‘D’ in Huntington’s Disease.
2. The Abstract needs to be critically revised, please add more information about the problem statement.
3. Please add more strong keywords.
4. Page 1, line 30-32: “Epigenetic mechanisms play essential roles in brain function through regulating gene expression [1-3]. Accumulating evidence implicates dynamic DNA methylation contributed to pathological process of brain [4, 5].” Please do not jump directly to DNA methylation. Make a connection between each paragraph.
5. Page 1, line 38: “cytosine at 5-position (Fig. 1A).” Please replace ‘Fig. 1A’ to ‘Figure 1A’
6. Page 3, line 71: “which share a homology catalytic domain at their C termini.” Please replace ‘C termini’ with ‘C terminal.’
7. Page 3: The whole introduction section looks general. Authors are advised to revise the introduction section carefully and add relevant data to support the problem statement and make a connection between each paragraph. There is no such information between 5mC and 5hmC during neural and brain development rather than general information about 5mC and 5hmC and how they formed. The introduction section also lacks information regarding the potential therapeutic approaches to target DNA methylation. Overall, an introduction needs a major revision.
8. Page 3: What is the research gap and novelty of the present study?
9. Page 5, line 142: “Furthermore, Dnmt3b is essential for de novo methylation…” Somewhere authors italic ‘de novo’ and somewhere not. Please remain consistent throughout the manuscript.
10. Page 5, line 178-179: “5mC / 5hmC are involved in neuronal l development…” What is neuronal l development?
11. Page 6, line 198: “CpG sites adjacent to ELVOL2 exhibited significant…” Please use the full form of the word when termed for the first time.
12. Page 6, 209-210: “Epigenetic aging rates based on DNA methylation were significantly correlated with race/ethnicity.” What is the reason for this?
13. Page 6, line 211-212: “Young male exhibited higher epigenetic aging rates than young woman, likely continue into old age.” This is an interesting point, please add more literature here to support this point. Also, add the reason behind it.
14. Page 6, line 216-217: “Using freshly isolated brain tissues from rhesus monkeys, we found that the 5hmC accumulated in four brain regions…” Please replace ‘we found’ with ‘they found.’
15. Please number each heading and subheading.
16. Page 7, line 254: “tau tangle aggregates.” Please replace ‘tau tangle’ with ‘Tau tangle.’
17. It is suggested to add a table explaining the genes involved in AD, PD, and HD and the therapeutic options to treat them.
18. Page 8, line 311: “and further impacting TNF-a transcriptional activation.” Please replace ‘TNF-a’ with ‘TNF-α.’
19. Page 8, line 313-314: “In addition, many other genes showed dysregulation of 5mC, including FANCC cg14115740, TNKS2 cg11963436, PGC-1α, and NOS2 by different screenings in PD cases.” Please italicize the names of genes and follow this trend throughout the manuscript.
20. Page 8, line 340-341: “A novel plant-derived nutraceutical, named atremorine, demonstrated neuroprotective effect on the dopaminergic neurons in the substantia nigra, possibly by improving DNA methylation and exerting epigenetic regulation.” Please add the mechanism of action of this drug with an illustration.
21. The authors have made very little discussion about the therapeutic options in DNA methylation. It is advised to add more data about therapeutic options with proper mechanisms and illustrations to catch the eyes of readers.
22. It is advised to remove the references from the conclusions and make it a single paragraph.
23. It is suggested to add two more headings, such as “Crosstalk of DNA methylation and other epigenetic mechanisms” and “Present challenges and future directions.”
24. The authors are advised to add a list of abbreviations.
25. Authors are advised to proofread the manuscript to overcome grammatical mistakes.
26. Authors are advised to revise headings and subheadings.
27. Most of the references are outdated; please revise them and add updated data.
Author Response
Reviewer 2
Comments and Suggestions for Authors
In this study, the author has studied “Dynamic regulation of DNA methylation and brain functions.” A lot of studies have already been carried out on a similar topic, and comprehensive data is available in the literature. Sentence-making is weak in this manuscript. The English language used in the manuscript needs major improvements as there are some punctuation and grammatical mistakes present throughout the manuscript. Experimental designs required more clarity. Moreover, research models are not discussed in an understandable manner. Repetition of lines is common, which reflects that the author needs a more comprehensive way of thinking.
Author Response: We thank the reviewer for the constructive comments. We have addressed the reviewer’s concerns point-by-point as follows:
Specific comments:
- Page 1, line 14-15: “Alzheimer’s disease (AD), Parkinson’s disease (AD) and Huntington’s Disease (HD).” Please remain consistent; either capitalize the word ‘D’ in all diseases or small the word ‘D’ in Huntington’s Disease.
Author Response: We appreciate the reviewer’s suggestion. We have revised it throughout the text, replacing ‘D’ with small ‘d’.
- The Abstract needs to be critically revised, please add more information about the problem statement.
Author Response: We thank for reviewer’s suggestion. The abstract has been revised as shown in the text as follows:
However, the mechanisms underlying aberrant DNA methylation in neurodegenerative diseases remain unclear. In this review, we provide an overview of the contribution of 5-methycytosine (5mC) and 5-hydroxylcytosine (5hmC) to brain development and aging with focus on the roles of dynamic 5mC and 5hmC changes in the pathogenesis of neurodegenerative diseases, particularly in Alzheimer’s disease (AD), Parkinson’s disease (PD) and Huntington’s disease (HD). Identification of aberrant DNA methylation sites could provide potential candidates for epigenetic-based diagnostic and therapeutic strategies for neurodegenerative diseases.
- Please add more strong keywords.
Author Response: We thank for reviewer’s suggestion. The keywords have been replaced as shown in the revised text as follows:
DNA methylation; transcriptional regulation; neurogenesis; development; aging; neurodegenerative diseases
- Page 1, line 30-32: “Epigenetic mechanisms play essential roles in brain function through regulating gene expression [1-3]. Accumulating evidence implicates dynamic DNA methylation contributed to pathological process of brain [4, 5].” Please do not jump directly to DNA methylation. Make a connection between each paragraph.
Author Response: We revised the above sentences as follows:
Epigenetic mechanisms play essential roles in brain function through regulating gene expression [1-3]. As a critical epigenetic modification, DNA methylation affects transcriptional activity by recruiting or inhibiting the binding of transcription factors to DNA. Many brain disorders display altered gene expression, and accumulating evidence implicates that dynamic DNA methylation contributed to altered gene expression and pathological process in the brain [4, 5].
- Page 1, line 38: “cytosine at 5-position (Fig. 1A).” Please replace ‘Fig. 1A’ to ‘Figure 1A’
Author Response: We have replaced it.
- Page 3, line 71: “which share a homology catalytic domain at their C termini.” Please replace ‘C termini’ with ‘C terminal.’
Author Response: We have corrected it.
- Page 3: The whole introduction section looks general. Authors are advised to revise the introduction section carefully and add relevant data to support the problem statement and make a connection between each paragraph. There is no such information between 5mC and 5hmC during neural and brain development rather than general information about 5mC and 5hmC and how they formed. The introduction section also lacks information regarding the potential therapeutic approaches to target DNA methylation. Overall, an introduction needs a major revision.
Author Response: We thank the reviewer for the comments. We introduced the generation of 5mC and 5hmC since several enzymes (methyltransferases and demethylases) participate in the process of DNA methylation and play roles in development and aging as well as in the pathogenesis of neurodegenerative diseases. We added description of the distribution and potential role of 5hmC in brain on Page 3 line 85-95. We also added the information about 5mC during neural and brain development (Page 2 line 59-64) and the potential therapeutic approaches to target DNA methylation (Page 4 line 105-118).
- Page 3: What is the research gap and novelty of the present study?
Author Response: Our review focus on the dynamic 5mC and 5hmC distribution and their roles in neurogenesis and aging with focus on important neurodegenerative diseases. The roles of dynamic 5mC and 5hmC in neurodegenerative diseases remain unclear. We have emphasized this in the revised introduction, which is also the focus of our review.
- Page 5, line 142: “Furthermore, Dnmt3b is essential for de novo methylation…” Somewhere authors italic ‘de novo’ and somewhere not. Please remain
Author Response: We have corrected it and kept the same way now.
- Page 5, line 178-179: “5mC / 5hmC are involved in neuronal l development…” What is neuronal l development?
Author Response: We would like to apologize for the confusion caused by typos. We have corrected it.
- Page 6, line 198: “CpG sites adjacent to ELVOL2 exhibited significant…” Please use the full form of the word when termed for the first time.
Author Response: We thank the reviewer for the comment. The full name of ELVOL2 has been added.
- Page 6, 209-210: “Epigenetic aging rates based on DNA methylation were significantly correlated with race/ethnicity.” What is the reason for this?
Author Response: This research analyzed multiple tissues of different races/ethnicities and found a correlation between DNA methylation and habitats that can influence aging. We have added this description to this part as follows:
Epigenetic aging rates based on DNA methylation were significantly correlated with race/ethnicity. Analyses of multiple tissues of different racial/ethnic groups suggested that habitat may be a noticeable factor affecting the aging acceleration [73].
- Page 6, line 211-212: “Young male exhibited higher epigenetic aging rates than young woman, likely continue into old age.” This is an interesting point, please add more literature here to support this point. Also, add the reason behind it.
Author Response: We have added more information about this point as follows:
Interestingly, the epigenetic clock of DNA methylation is also correlated with gender. Young male exhibited higher epigenetic aging rates than young woman, likely continue into old age, especially in brain tissue [74-76]. But the cerebellum ages more slowly, and show no significant sex-specific effects [73]. Accumulation of DNA damage and DNA repair mechanisms are associated with aging and age-related genomic instability in males, in other words, the efficacy of DNA repair mechanisms may differ between males and females [77,78]. Another explanation is that removing androgen could slow down the rate of aging [79]. However, this difference between males and females has been difficult to verify in animal models. Therefore, it is necessary to analyze the differences epigenetic aging rates in males and females throughout the lifespan.
- Page 6, line 216-217: “Using freshly isolated brain tissues from rhesus monkeys, we found that the 5hmC accumulated in four brain regions…” Please replace ‘we found’ with ‘they found.’
Author Response: We have corrected it.
- Please number each heading and subheading.
Author Response: We have added number to each heading.
- Page 7, line 254: “tau tangle aggregates.” Please replace ‘tau tangle’ with ‘Tau tangle.’
Author Response: We have replaced it.
- It is suggested to add a table explaining the genes involved in AD, PD, and HD and the therapeutic options to treat them.
Author Response: We summarized them in the Table 1.
- Page 8, line 311: “and further impacting TNF-a transcriptional activation.” Please replace ‘TNF-a’ with ‘TNF-α.’
Author Response: We have replaced it.
- Page 8, line 313-314: “In addition, many other genes showed dysregulation of 5mC, including FANCC cg14115740, TNKS2 cg11963436, PGC-1α, and NOS2 by different screenings in PD cases.” Please italicize the names of genes and follow this trend throughout the manuscript.
Author Response: We have checked and italicized the names of genes.
- Page 8, line 340-341: “A novel plant-derived nutraceutical, named atremorine, demonstrated neuroprotective effect on the dopaminergic neurons in the substantia nigra, possibly by improving DNA methylation and exerting epigenetic regulation.” Please add the mechanism of action of this drug with an illustration.
Author Response: The potential mechanism of atremorine has been added in the revised text as follows:
A novel plant-derived biopharmaceutical compound, named atremorine, reduced the production of tyrosine hydroxylase in the substantia nigra, blocked microglia activation, and prevented neuronal death. It has an neuroprotective effect on the dopaminergic neurons in the substantia nigra, possibly by improving DNA methylation and exerting epigenetic regulation [136]. For example, atremorine prevented 1-methyl-4-phenyl-1,2,3,6-tetrahydropyridine (MPTP) from inducing dopaminergic neurodegeneration in the substantia nigra and microglia activation and neurotoxicity in the nigra-striatal region. Mice with MPTP-induced neurodegeneration also had improved motor and cognition function [136].
- The authors have made very little discussion about the therapeutic options in DNA methylation. It is advised to add more data about therapeutic options with proper mechanisms and illustrations to catch the eyes of readers.
Author Response: More discussion about therapeutic options has been added as follows:
Formaldehyde is the main precursor of most complex organic molecules [97]. Increased concentration of formaldehyde in the brain can lead to neurological dysfunction and even degeneration [98,99]. The process of DNA demethylation is accompanied by the generation of formaldehyde [100]. Early AD-like changes occurred in mice when they were exposed to excessive formaldehyde daily for one week [101]. After chronic intracerebroventricular injection of formaldehyde in rhesus monkeys, the animals exhibited Aβ neuritic-like plaques, neurofibrillary tangle-like structure, increased phosphorylation tau protein, neuronal loss in the hippocampus, entorhinal cortex, and prefrontal cortex [102]. Interestingly, active DNA demethylation process was also induced upon formaldehyde administration [103]. Injection of excessive formaldehyde into the hippocampus of healthy adult rats mimics abnormalities in learning and cognition. It may be due to the reduction in global DNA methylation levels caused by the inhibition of DNMT1 and DNMT3a activity through modification of their cystine residues by excess formaldehyde [104]. Excessive physiological levels of formaldehyde interfered with overall DNA methylation reduction in AD patients [104]. Therefore, excessive formaldehyde exposure from environment led to abnormal DNA methylation and may be one of the risk factors for AD [105].DNA methylation editing has been used as a novel treatment for a variety of diseases. Abnormal CpG methylation of the APP led to abnormal gene expression of the APP in the brains of AD patients. APP hypermethylation can be achieved by editing system to fuse Tet1 or Dnmt3a with dCas9 to effectively inhibit APP in primary neurons and in vivo mouse brains [106-108].
- It is advised to remove the references from the conclusions and make it a single paragraph.
Author Response: We reorganized this part as ‘Present challenges and Perspectives’ and ‘Conclusion’ in which there is no reference.
- It is suggested to add two more headings, such as “Crosstalk of DNA methylation and other epigenetic mechanisms” and “Present challenges and future directions.”
Author Response: We have added these two parts.
- The authors are advised to add a list of abbreviations.
Author Response: We have added a list of abbreviations.
- Authors are advised to proofread the manuscript to overcome grammatical mistakes.
Author Response: We have checked English carefully and corrected some grammar errors.
- Authors are advised to revise headings and subheadings.
Author Response: We have revised the headings and subheadings accordingly.
- Most of the references are outdated; please revise them and add updated data.
Author Response: We have carefully checked references and updated some references.

Reviewer 3 Report
The authors presented a nice review of recent advancements of DNA cytosine methylation in brain development, aging and neurodegenerateive diseases. The manuscript is qualified to be pulished on Biology with minor revisions.
1) In Figure 1B, text resolution and contrast needs to improve.
2) Line 202 the question ‘An important question is whether DNA methylation functions as a footprint mark or driver of aging?’ was not discussed clearly in the following context. The examples given in the paragraph showed correlation between DNA methylation and aging, but whether there is causal effect of methylation to aging?
3) Line 213 change ‘f5hmC’ to ‘5hmC’.
Author Response
Reviewer 3
Comments and Suggestions for Authors
The authors presented a nice review of recent advancements of DNA cytosine methylation in brain development, aging and neurodegenerative diseases. The manuscript is qualified to be published on Biology with minor revisions.
Author Response: We thank the reviewer for the appreciation of our work. We have addressed the reviewer’s concerns point-by-point as follows:
- In Figure 1B, text resolution and contrast needs to improve.
Author Response: The Figure 1B at high resolution has been included to replace the previous one.
- Line 202 the question ‘An important question is whether DNA methylation functions as a footprint mark or driver of aging?’ was not discussed clearly in the following context. The examples given in the paragraph showed correlation between DNA methylation and aging, but whether there is causal effect of methylation to aging?
Author Response: This is an interesting issue. In the revised text, we have added the following discussion:
A number of studies have tried to explore the relationship between DNA methylation and aging. Currently, DNA methylation is regarded as a "ruler" measuring the aging of organisms”. However, it is well known that aging is a multifactorial process, and gene expression is not completely regulated by DNA methylation.
3) Line 213 change ‘f5hmC’ to ‘5hmC’.
Author Response: We would like to apologize for the confusion caused by typos. We have corrected it.

Reviewer 4 Report
1) The topic is interesting, however, the authors skimmed through the titles and did not pay attention to the mechanisms. Moreover, the abstract was written very briefly and the most important results regarding the association of DNA methylation with brain disorders are not mentioned.
2) DNA methylation is not the only factor in brain aging. The most important factor is telomere shortening and histone PTM patterns. It would be more informative to add a Venn diagram in order to show the weight of each of the aftermentioned factors in causing brain aging.
3) It is obvious that alterations in methylation patterns is contributed to AD and PD patients. However, it is important to discuss the mechanism behind them and summarized them in figures.
Overall, the information in the review manuscript is similar to other reviews, nothing is new.
Author Response
Reviewer 4
Comments and Suggestions for Authors
- The topic is interesting, however, the authors skimmed through the titles and did not pay attention to the mechanisms. Moreover, the abstract was written very briefly and the most important results regarding the association of DNA methylation with brain disorders are not mentioned.
Author Response: We thank the reviewer for the comment. One of the big challenges of investigation of DNA methylation in the brain is to unravel precise mechanisms how DNA methylation regulates the genome instability and gene expression. There are not many literatures clearly defining the mechanisms and most of studies describe the correlation or relationship between DNA methylation and brain function. Thus, we tried to discuss possible mechanisms behind the observed aberrant changes of DNA methylation in the brain, particularly in neurodegenerative diseases. We have revised the abstract as follows:
DNA cytosine methylation is a principal epigenetic mechanism underlying transcription during development and aging. Growing evidence supports that DNA methylation plays a critical role in brain function, including neurogenesis, neuronal differentiation, synaptogenesis, learning, and memory. However, the mechanisms underlying aberrant DNA methylation in neurodegenerative diseases remain unclear. In this review, we provide an overview of the contribution of 5-methycytosine (5mC) and 5-hydroxylcytosine (5hmC) to brain development and aging with focus on the roles of dynamic 5mC and 5hmC changes in the pathogenesis of neurodegenerative diseases, particularly in Alzheimer’s disease (AD), Parkinson’s disease (PD) and Huntington’s disease (HD). Identification of aberrant DNA methylation sites could provide potential candidates for epigenetic-based diagnostic and therapeutic strategies for neurodegenerative diseases.
- DNA methylation is not the only factor in brain aging. The most important factor is telomere shortening and histone PTM patterns. It would be more informative to add a Venn diagram in order to show the weight of each of the aftermentioned factors in causing brain aging.
Author Response: We thank the reviewer for the suggestion. A Venn diagram has been added as Figure 3.
- It is obvious that alterations in methylation patterns is contributed to AD and PD patients. However, it is important to discuss the mechanism behind them and summarized them in figures 3.
Author Response: We thank the reviewer for the suggestion. We summarized the potential mechanisms in Figure 4-6.

Round 2
Reviewer 2 Report
The authors have carefully addressed all the comments. So, the manuscript should be accepted in its present form.
Reviewer 4 Report
The authors provided my answers.